# Unique Jurassic Ovaries Shed a New Light on the Nature of Carpels

**DOI:** 10.3390/plants13162239

**Published:** 2024-08-12

**Authors:** Qiang Fu, Jie Sun, Shaolin Zheng, Xin Wang

**Affiliations:** 1State Key Laboratory of Palaeobiology and Stratigraphy, Nanjing Institute of Geology and Palaeontology and CAS Center for Excellence in Life and Paleoenvironment, Chinese Academy of Sciences, Nanjing 210008, China; qiangfu@nigpas.ac.cn; 2University of Chinese Academy of Sciences (UCAS), Nanjing 211135, China; 3Shaanxi Key Laboratory of Early Life and Environments, State Key Laboratory of Continental Dynamics, Department of Geology, Northwest University, Xi’an 710069, China; sj_10242016@163.com; 4The Shenyang Center of Geological Survey and Institute of Geology and Mineral Resources, Ministry of National Land and Resources, Shenyang 110034, China

**Keywords:** *Xenofructus*, fruits, ovary, free central placentation, evolution, angiosperms, Jurassic

## Abstract

Enclosed ovules are a reproductive feature restricted to angiosperms. Although this feature can be used as a criterion for identifying fossil angiosperms, how ovules are enclosed and the nature of the placenta are still foci of debates. A reason underlying these controversies is the lack of reproductive organ fossils shedding light on these issues. These controversies hinder a clear understanding of angiosperm evolution and systematics. Here, we report a new fossil ovary, *Xenofructus dabuensis* gen. et sp. nov, from the Middle Jurassic of Liaoning, China. Our fossil clearly demonstrates the existence of ovules in *Xenofructus* that has a free central placentation. This new feature implies that a placenta in angiosperm gynoecia is homologous to an ovule/seed-bearing axis, and free central placentation is one of the early developed placentations. This discovery is apparently at odds with the current understanding of placentation and its evolution. Apparently, the understanding of angiosperms and their gynoecia should be updated with newly available palaeobotanical data.

## 1. Introduction

Angiosperms are the single most diversified plant group and they are closely related with the well-being of the humans. Ovaries are a feature idiosyncratic to angiosperms. Formerly, a conduplicate carpel with unilocular ovary was taken as the most ancestral type of placentation and all other types were derived from this type through various modifications [1,2]. However, this thought has been disfavored by fossils of early angiosperms uncovered in the past decades ([3,4,5,6,7,8,9], to name a few). A recently raised hypothesis stated that a carpel is a composite organ derived from an ovule-bearing axis and a subtending and enclosing leaf [10,11,12]. The latest progress in this research includes the following: (1) recognizing the branch-terminal position of seeds in gymnosperms [13,14], and (2) treating ovule and carpel (wall) separately, which become tightly synorganized, in extant angiosperms [15]. Although these moves are in the right direction, the controversy remains on ovule/seed position in ovaries of early angiosperms, including famous *Archaefructus* [16,17,18,19] and *Archaeanthus* [20,21]. In these debates, the seeds were interpreted as borne either on the adaxial or the abaxial margin (two opposite positions) of the ovaries/ovaries, and neither of the confronting schools could convince the other. Although there seems to be a consensus on the evolution of placentation in angiosperms, this consensus apparently lacks the testing and favoring from fossil evidence. The stalemate between these two conflicting parties cannot be broken until an independent informative fossil occurs. Here, we report a new fruit/ovary, *Xenofructus dabuensis* gen. et sp. nov, from the Middle Jurassic of Liaoning, China. Surprisingly, these Jurassic ovaries clearly demonstrate the existence of free central placentation in early angiosperms. It appears that this new fossil provides a new insight into the evolution of placentation in angiosperms.

## 2. Materials and Methods

The specimen was collected from an outcrop of the upper member of the Dabu Formation in the Tianshifu Coal Mine (41°14′58″ N, 124°23′03″ E) in 1978 by Dr. Shaolin Zheng and Dr. Wu Zhang. The specimen bearing the fossil plants is dark-colored siltstone, lacking good lamination. There are two similar ovaries preserved on the surface of the same block. Previously reported fossil plants from the upper member of the Dabu Formation include the following: *Equisetum beani, E. laterale, Neocalamites hoerensis, Todites williamsoni, Dicksonia concinna, Eboracia lobifolia, Clathropteris pekingensis, Cladophlebis asiatica, C.* sp., *Coniopteris simplex, Pterophyllum propinquum, P. angustum, P.* sp., *Ctenis kaneharai, Williamsoniella buracovae, W. minima*, ***W. dabuensis****, Nilssonia tenuinervis, Ginkgo concinna, G.* sp., *Czekanowskia* sp., *Baiera gracilis, B. furcata, Sphenobaiera gyron, Phoenicopsis augustifolia, Sphenarion latifolia,* “*Elatidies*” *chinensis, Storgaardia mentoukouensis*, *Lindleycladus lanceolatus*, *Carpolithus* sp., and *Raticites* sp. Other than these taxa, the fossil plants from the Dabu Formation include *Lycopodites falcatus, Neocalamites carrerei, Annulariopsis* sp., *A.* sp., *Marattiopsis hoerensis, M. muensteri*, *Coniopteris hymenophylloides, C. kuandianensis, C. karatiubensis, Kuandiania crassicaulis*, *Clathropteris meniscioides, Hausmannia* cf. *dentata*, *Todites denticulata, Dicksonia kendalli*, *Cladophlebis raciborskii, C. argutula, C. shansiensis, C. fukiensis, C. haiburnensis, C. nebbensis*, *Pterophyllum issykkulense, Nilsonia mosserai, N.* aff. *acuminata*, *N.* cf. *compta*, *N. orientalis, N. mosserayi, Anomozamites* cf. *major*, *A. inconstans, A.* spp., *Pterophyllum aequale, P. ctenoites*, *P. nathorsti*, *Ctenis lingyuanensis*, *Pityophyllum nordenskioldi, P. longifolium, Tyrmia grandifolia, Baiera asadai*, *B.* cf. *lindleyana*, *Ginkgo beijingensis, G. obrutschewi*, *G. manchurica, G. sibirica*, *Sphenobaiera huangi*, *Czekanowskia rigida*, *Phoenicopsis* aff. *speciosa, Elatidies ovalis, Elatocladus* (*Cephalotaxopsis*?) *angustifolius*, *E. lindongensis, Ferganiella* cf. *podozamioides, Solenites vimineus, Leptostrobus laxifolia,* and *Isostrobus heeri* [22,23]. The fossil flora is comparable to the *Coniopteris-Phoenicopsis* flora, and the fossiliferous strata are thought to belong to the early Middle Jurassic as it shares 17 species with the typical Middle Jurassic Yorkshire Flora [22,23].

The whole specimen was photographed using a Huawei digital camera, and details of the fossil were photographed using a Nikon SMZ1500 stereomicroscope at the Nanjing Institute of Geology and Paleontology, Nanjing, China. The specimen was investigated on a Phoenix v|tome|x m scanner using a voltage of 190 Kv and a current of 140 μA at the State Key Laboratory of Continental Dynamics (Northwest University), Xi’an, Shaanxi Province, China. The specimen was mounted on the stage using self-adhesive tape. In total, 2000 projections were obtained. The dataset had a resolution of 74.56 μm. Three-dimensional reconstructions were generated using VG Studio 3.2. All images were saved in TIFF or JPEG format. The brightness and contrast of the images were adjusted as a whole. All figures were organized for publishing using Photoshop 7.0 software.

## 3. Results


**Angiosperms**



***Xenofructus* gen. nov**


**Synonym**: *Williamsoniella* Zheng and Zhang 1990

**Type species**: *Xenofructus dabuensis* gen. et sp. nov

**Etymology**: *Xeno–* for unexpected morphology and occurrence of the fossil, *–fructus* is the Latin word for fruit.

**Generic diagnosis**: Fruit/ovary with a short stalk, straight on abaxial, convex on adaxial. Abaxial ridge robust. Placenta free from both abaxial and adaxial margins of fruit/ovary. Ovules enclosed in fruit/ovary, in cluster, sickle-shaped.


**Remarks:**


The specimens studied here were previously identified as *Williamsoniella dabuensis* by Zheng and Zhang in 1990 [22], and were placed in Bennettitales. Indeed, the general morphology did resemble what shown in Text-fig. 3 of Harris [24]. However, the features of our specimen appear to fall out of the morphological scope of Bennettitales, which are expected to have a conic receptacle bearing numerous seeds dispersed among interseminal scales or pollen sacs on the adaxial surface of microsporophylls [24,25]. Forms and variations of ovules in our specimen suggest that they are not pollen sacs, as suggested by Zheng and Zhang (1990), which are expected to be more uniform in morphology. Therefore, we re-studied the specimen and revealed some new features. 

The ovary of *Xenofructus dabuensis* gen. et sp. nov is unique in its placenta position (internal in the ovary, positioned between the adaxial and abaxial margins of the ovary) and sickle-shaped ovules borne in clusters on a branch. The combination of these features has never been seen in any previously reported fossil taxa, thus justifying a new genus.


***Xenofructus dabuensis* gen. et sp. nov**


(Figure 1 and Figure 2) 

**Synonym**: *Williamsoniella dabuensis* Zheng and Zhang 1990, Plate V, Figure 7A, page 220 [22]

**Diagnosis**: The same as that of the genus.

**Description**: The specimens studied here include two isolated fruits/ovaries in parallel oriented on the surface of the same block (Figure 1 and Figure 2). Both ovaries are preserved as coalified compressions (Figure 1a,b). The general morphology of these two ovaries agrees with each other, although there is a slight difference in dimension (Figure 1a,b,g). One of the ovaries (Ovary A) measures 19 mm long and 8.4 mm wide (Figure 1a), while the other (Ovary B) measures 21 mm long and 6.9 mm wide (Figure 1b). Ovary A has a stalk 4.7 mm long and 2.5 mm wide, gradually transitional to the base of the ovary (Figure 1a and Figure 2a). Ovary B has a stalk 5.6 mm long and 1.8 mm wide, gradually transitional to the base of the ovary (Figure 1b and Figure 2b). Due to differences in breaking, different features are shown in these two ovaries, respectively (Figure 1a–f and Figure 2a,b). The foreground part of Ovary A is almost fully removed, exposing the internal cluster of ovules and robust abaxial ridge (Figure 1a,g,h and Figure 2a), while the foreground part of Ovary B is partially removed, with the clustered ovules (placenta) partially covered by the ovary wall (Figure 1b,c,e and Figure 2b). The internal position of the placenta (cluster of ovules) within the ovary is suggested by the presence of the stubs of the ovary wall (Figure 1b,c,e). The placenta is a central axis bearing several ovules (Figure 1a–c and Figure 2a,b). The ovules are sessile, with different orientations and morphologies, depending on their positions in the placenta (Figure 1a–c,f and Figure 2a,b). The basal ovules tend to be corpulent and round in shape, up to 1.9 mm long and 0.95 mm thick (Figure 1c), while the distal ones tend to be more slender and sickle-shaped, up to 2.5 mm long and 1.1 mm thick (Figure 1c,f). There appears to be some trace of a micropyle at the distal of a seed, suggestive of orthotropous ovules (Figure 1c,f).

**Holotype specimen**: KP10-18A.

**Etymology**: *dabuensis–*, species epithet conserved from *Williamsoniella dabuensis* Zheng and Zhang, former name of the specimen.

**Type locality**: Tianshifu Coal Mine, Liaoning, China.

**Stratigraphic horizon**: The Dabu Formation (the Middle Jurassic, >164 Ma).

**Depository**: The Shenyang Center of Geological Survey and Institute of Geology and Mineral Resources, Ministry of National Land and Resources, Shenyang, China.

## 4. Discussion

**Angiosperm affinity.** The term “angiosperms” was coined by German botanist Paul Hermann (1646–1695) in 1690. He combined two Greek words, *angeion* (vessel) and *spermos* (seed), into a new word “angiosperm” designating “plant with seeds contained in a protective vessel” (www.etymonline.com). This definition has been accepted widely by many leading botanists [26]. For example, the so-called “First Flower” *Archaefructus liaoningensis* was recognized following this definition. Wang [6,7,8,9,10,27,28,29,30,31,32,33,34,35,36] and others [16,17,18] have applied this criterion, explicitly or implicitly, identifying various early angiosperms. Thus, in this paper, we adopt this criterion for angiosperms to identify our fossil plant, *Xenofructus* gen. nov. 

As ovules are supposed to be enclosed in angiosperm ovaries, angiosperm ovules are not to be visible to the naked eye unless the enclosing ovary wall is broken or transparent. This makes proving the existence of enclosed ovules in a fossil plant a great challenge for palaeobotanists, especially when there is only one specimen of fossil fruit/ovary available for study. We are lucky in that there are two ovaries preserved side by side in the same specimen and these two ovaries are preserved in slightly different statuses (Figure 1a,b). Ovary A in Figure 1a is a broken ovary; namely, the foreground ovary wall is almost fully removed (Figure 1a,g,h), exposing the ovules formerly covered in the ovary and thus allowing us to view all clustered ovules. These ovules are clustered along an axis, forming an equivalent of a placenta. The counterpart of this placenta is only partially seen in Ovary B (Figure 1b,c), as the placenta in this ovary is partially covered (and eclipsed) by the ovary wall (Figure 1b,c). Comparing these two ovaries suggests that the placentae (including the ovules on them) of *Xenofructus* are sandwiched between two ovary walls and thus fully enclosed in the formerly integral ovaries, satisfying the criterion of angiosperms. 

The presence of an ovary wall in *Xenofructus* is suggested by the partially preserved placenta in Ovary B. Comparing Figure 1a–c, it is easy to see that there are fewer ovules in Figure 1b,c than in Figure 1a, and that the ovules in both ovaries differ from each other in size, outline, orientations, and number. These differences are due to the partial removal and partial preservation of the ovary wall in Figure 1b,c. The presence of the ovary wall is further suggested by Figure 1e, which shows a detailed view of the basal portion of Ovary B in Figure 1b, in which the fallen-off ovary wall left a clear stub. Both Figure 1e and Figure 1b,c indicate that the ovary wall covers both basal and upper parts of the ovary. Actually, the presence of the ovary wall is obvious on the adaxial and abaxial margins if a careful observation is made: in Figure 1b,c, the dark shadow of the ovary wall, although thin, on the left is clearly shown in Ovary B. A similar residue of ovary wall or epidermis is also visible in Ovary A: a robust straight ridge (abaxial margin) of Ovary A (Figure 1a,d,g,h) is covered by an additional thin layer. It is not pure imagination that the ovary wall was integral and covered the whole ovary, although the current preservation cannot show this. 

Although the age of *Xenofructus* is the Middle Jurassic and the so-called mainstream thinking rejected all pre-Cretaceous angiosperms, Figure 1a–j and Figure 2a,b have demonstrated clearly that the ovules in *Xenofructus* are sandwiched and enclosed in the ovaries. These images of *Xenofructus* and the above-adopted definition pin down that *Xenofructus* is a bona fide angiosperm.

**Two alternative interpretations.** According to the definitions given for various types of placentation in angiosperms in a survey by Shivaprakash et al. [2], *Xenofructus* should be categorized as an atypical “free central placentation”, as it has a unilocular gynoecium and a central column (bearing ovules) in the ovary (page 2 of [2]). However, *Xenofructus* is different from a typical free central placentation in terms of carpel number and general morphology. A usually referred free central placentation comprises three or more carpels, the ovary is of radial symmetry, and the placenta is centrally positioned when viewed from the top. However, *Xenofructus* comprises only one carpel, is very likely of bilateral symmetry and its placenta is skewed to one side when viewed from the top. Such distinctions force us to distinguish *Xenofructus* from typical free central placentation; therefore, we prefer to term it as having an “atypical free central placentation”, although it fits perfectly in the definition of “free central placentation” given by Shivaprakash et al. [2].

Although atypical, *Xenofructus*’ placentation does shed some light on the evolution of placentation in angiosperms. The earliest record of free central placentation is that documented in *Xingxueanthus*, in which ovules were interpreted arranged around a central column inside the ovary bearing a very short style [5]. The Jurassic age of *Xingxueanthus* indeed makes it a surprise for many, as free central placentation had been interpreted as a derived (not primitive or ancestral) type of placentation [2,37,38]. Its early occurrence challenges the mainstream thinking about angiosperm evolution. Now *Xenofructus* with free central placentation (although atypical) from the Middle Jurassic challenges the mainstream thinking on angiosperm evolution again. Maybe now is a decent time to re-evaluate the long-held concepts about angiosperm evolution. Comparing *Xingxueanthus* and *Xenofructus* of distinct morphologies, the shared features between them are restricted to (1) the Jurassic age, and (2) ovules borne on a column/branch. Focusing on the latter feature, it is easy to draw a conclusion: ovules are borne on branches. This conclusion contradicts Goethe’s dictum “Alles is Blatt” (leaf is everything), which is a foundation for angiosperm systematics that placed Magnoliales as the basalmost angiosperm group since 1907. Actually, rejections of this wrong assumption have been presented and have been constantly increasing. First, it is well-known that the earliest land plants have no leaf at all and leaf is derived from branches [39,40,41,42]. Second, all early seeds are on the termini of branches [43,44]. Third, at least some Mesozoic seeds are borne on branch termini [13,14]. Fourth, seeds in Cycadales [45,46], Ginkgoales [10], and Gnetales [47,48] are borne on branches, not leaves, while the seed-bearing scales in conifers are metamorphosed branches [49,50]. Now, new evidence from *Xenofructus* is appended to these existing challenging data. Furthermore, studies on function genes have long suggested that ovules are borne on branches that later were recruited onto the carpel margins [51,52,53,54]. Such a change in perspectives on seed-bearing structures makes Endress’ minute adjusting of his interpretation of carpels understandable. Unlike formerly, Endress tended to separate a former atomic carpel into two parts, ovule and carpel, although he refused to tell the reader on which part the seeds/ovules are borne. Taking all the above into consideration, we tend to interpret a placenta as a branch bearing ovules/seeds. This interpretation has at least one advantage: the placenta is comparable and homologous with an ovule/seed-bearing structure in gymnosperms. Solving the problem of placenta homology is a big step in our deciphering the origin of angiosperms and their gynoecia (androecia are much less challenging).

Another alternative interpretation cannot be eliminated completely. *Xenofructus* with atypical free central placentation may be otherwise compared with follicles with marginal placentation. Compared with typical follicles, the most intriguing feature of *Xenofructus* may be its placenta positioning. Unlike previously reported fossil plants, including *Archaefructus* [16,17,18,19], *Sinocarpus* [55,56], and *Archaeanthus* [20,21], in which the seeds/ovules were interpreted as borne either on the adaxial or abaxial margin of fruits (frequently called follicles) (Figure 3), the ovules in *Xenofructus* are clustered around an axis independent of both adaxial and abaxial margins. Just like seed-bearing branches in gymnosperms, the occurrence of such a placenta in *Xenofructus* (1) declares the independence of placenta in carpels/gynoecia of angiosperms, (2) potentially represents an ancestral status of the placenta, suggesting the derivedness of the frequently seen adaxial or abaxial position of ovules in an ovary. This conclusion is in line with the study on *Michelia* [12] and the generalization given by Wang [10]. Most importantly, studies on gene function have long given similar implications [51,52,53,54], which, unfortunately, have been largely ignored by many botanists. Such a consensus among paleobotany, plant morphology, and molecular studies seems to indicate that we are approaching the truth.

**Theoretical consideration.** Although the conclusion on the evolution of placentation types of Shivaprakash et al. [2], just like previous ones [37,38], points to marginal placentation as the basalmost one, we tend take it as tentative, because (1) at least two Jurassic fossil plants have free central placentation, contradicting the conclusion of Shivaprakash et al., (2) the conclusion of Shivaprakash et al. [2] is drawn based on one assumption: the APG system reflects the truth of angiosperm systematics, which unfortunately is hard to disentangle its link with the Mangoliales-centered angiosperm system that now is found to be groundless. Assuming so, it would be hard to conclude otherwise. Considering expectations of theories based only on extant angiosperms are rarely honored by early fossil angiosperms, we prefer to test the conclusion of Shivaprakash et al. more with independent fossil evidence before adopting it.

**Development in fossils.** Ovules of *Xenofructus* vary in their morphology, even in the same ovary (Figure 1a–c,f). Such variation is rather expected, as ovules borne on the same placenta may be in their different developmental stages, as seen in extant *Nymphaea* [58]. For example, three adjacent ovules of Ovary B are of different shapes (Figure 1c). The lowermost one is more corpulent and round in shape, the uppermost one is apparently more slender and smaller, sickle-shaped, while the middle one is intermediate between these two. Such variation suggests an acropetal development of the ovules in *Xenofructus*. It is noteworthy that all ovules are consistently sickle-shaped. Not alone, a similar variation of ovules can be seen in Ovary A (Figure 1a). It is important that there seems to be an invagination at the tip of the seed/ovule (Figure 1f). This conjures to micropyle, a feature frequently seen at the tip of an ovule, rejecting former interpretation of these ovules as pollen sacs. 

**Origin time of angiosperms** There are increasing independent molecular datings suggesting a Jurassic or even Triassic origin of angiosperms [59,60,61,62,63,64]. A recent study using numerous genes of many taxa in angiosperms and 62 fossil calibrations [63,65] suggested that there is a “Jurassic Gap” for angiosperms. This conclusion is further strengthened by the latest phylogenomic study on 353 gene sets in almost 800 genera of angiosperms carried out by 279 scientists from 138 institutions in 27 countries [64]. Now, our *Xenofructus* represents a Middle Jurassic (>164 Ma) angiosperm, filling the so-called “Jurassic Gap”. It appears that the impression of the “Jurassic Gap” is either due to the lack of knowledge on Jurassic angiosperms or due to intentionally ignoring Jurassic angiosperms, just as Herendeen et al. [66] did. We cannot know the truth until we study plant fossils carefully. Actually, there have been enough studies suggestive of pre-Cretaceous angiosperms [6,7,8,9,10,34,36,67,68,69,70,71,72,73,74]. We should take them into future consideration.

**Perspectives** We still do not know anything about the whole plant of *Xenofructus* yet. The occurrence of two *Xenofructus* ovaries on the surface of the same specimen seems to suggest that *Xenofructus* proliferated in the local flora. These ovaries are parallel in orientation, suggesting either that they belonged to the same plant and were connected to each other by an unpreserved part or that they represented two of many ovaries in the fossil flora, implying many unseen ovaries awaiting further digging in the fossil flora. We expect future studies will shed more light on *Xenofructus* and other angiosperms that lived in the Jurassic.

Previous studies on megafossils and palynology of the Dabu Formation showed that the fossil flora bearing *Xenofrucuts* was dominated by Cycadales (28.57%), ferns (27.27%) and Ginkgoales (24.68%), followed by Coniferales (11.69%), Equisetales (6.49%), and Lycopodiales (1.3%) [23]. These plants constitute a forest dominated by trees of Ginkgoales, Czekanowskiales, and Coniferales, with flourished bushes of Cycadales and Bennettitales, and an understory of ferns [23]. We hope this information can be helpful for future studies on the ecological background of angiosperm evolution. 

## 5. Conclusions

*Xenofructus* represents a new taxon of Jurassic angiosperms. Its early age and unique morphology not only confirm the truthful occurrence of angiosperms in the pre-Cretaceous age but also shed new light on the homology of placenta in angiosperms as well as the origin of angiosperms.

## Figures and Tables

**Figure 1 plants-13-02239-f001:**
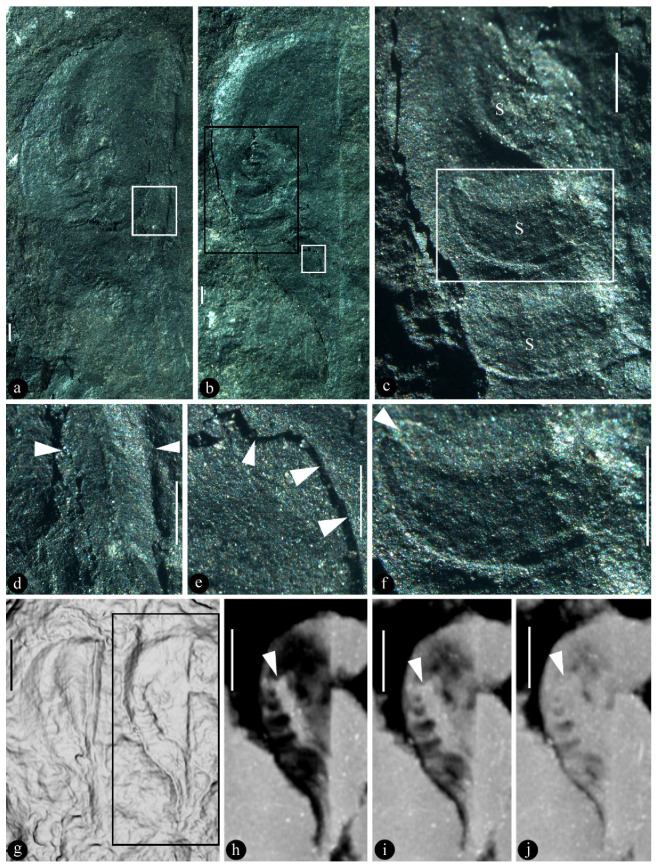
*Xenofructus dabuensis* gen. et sp. nov and its details. Specimen number KP10-18A. (**a**). Holotype, showing the profile of Ovary A and its seeds/ovules exposed after the foreground ovary is removed. Scale bar = 1 mm. (**b**). Ovary B and its internal ovules partially exposed and partially covered by the broken ovary wall. Scale bar = 1 mm. (**c**). Detailed view of the black rectangle in (**b**), showing three ovules (s) of variable shapes and ovary wall (upper right) covering other ovules. Scale bar = 1 mm. (**d**). Detailed view of the rectangle in (**a**), showing robust abaxial margin (between the triangles) of the ovary. Scale bar = 1 mm. (**e**). Broken ovary wall (upper right) and exposed internal of the ovary (lower left), and the boundary in between (triangles), enlarged from the white rectangle in (**b**). Scale bar = 0.5 mm. (**f**). A sickle-shaped ovule with a distal invagination (possible micropyle, triangle), enlarged from the rectangle in (**c**). Scale bar = 1 mm. (**g**). A micro-CT rendering of the specimen showing two ovaries parallel deployed on the surface of the specimen. (**h**–**j**). Serial sections of Ovary B, showing the ovule-bearing placenta (triangles) independent from both margins of the ovary. Scale bar = 2 mm.

**Figure 2 plants-13-02239-f002:**
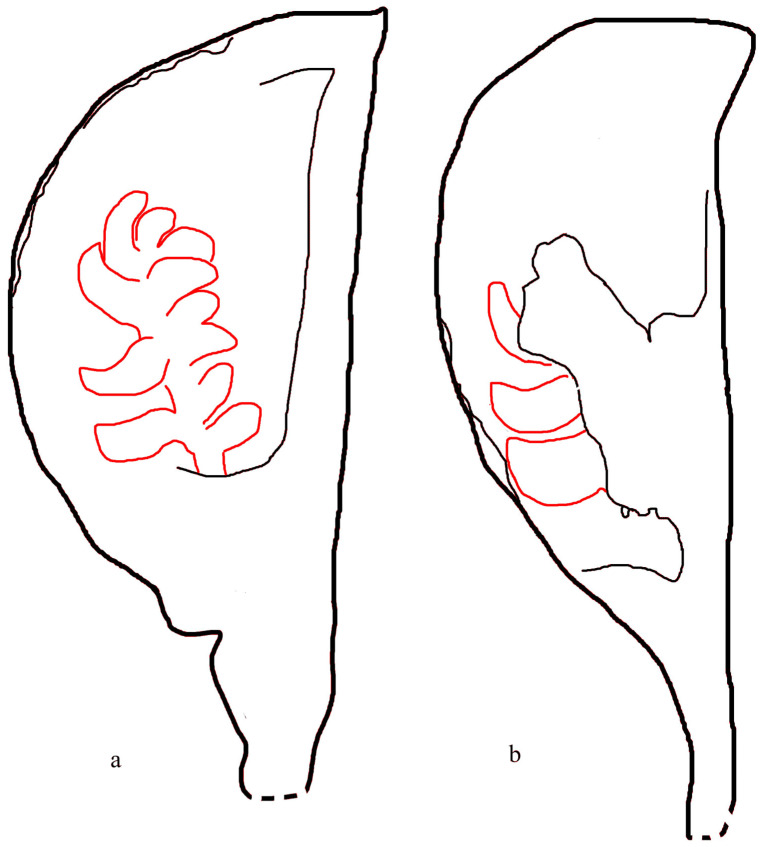
Sketches of two *Xenofructus* ovaries. The ovary outline is in black, while the seed outline is in red. (**a**). The ovary in Figure 1a. (**b**). The ovary in Figure 1b.

**Figure 3 plants-13-02239-f003:**
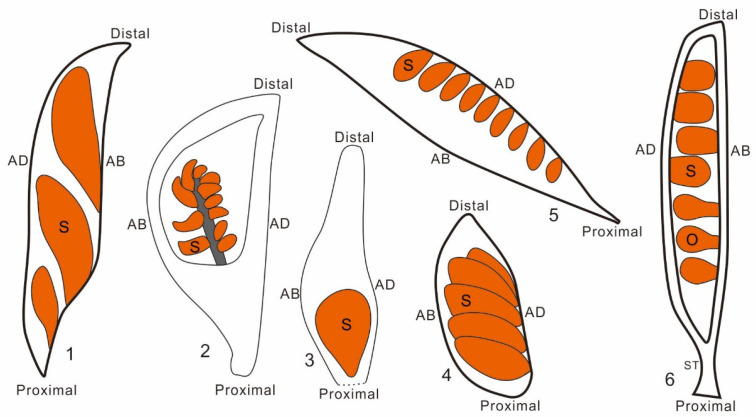
Several representatives of early angiosperms, showing variable placentations in the fruits/ovaries of early angiosperms. Seeds (S)/ovules (O) are inserted either along the adaxial (AD), abaxial (AB), or both margins of a fruit/ovary, or are basally fixed in a fruit/ovary. 1. *Archaefructus*, according to Wang and Zheng [19]; 2. *Xenofructus*; 3. *Qingganninginfructus*, according to Han et.al. [36]; 4. *Gansufructus*, according to Du et al. [57]; 5. *Sinocarpus*, according to Leng and Friis [55]; 6. *Archaeanthus*, according to Wang [20].

## Data Availability

All data generated or analyzed during this study are included in this published article.

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
