# Peer review of "Unique Jurassic Ovaries Shed a New Light on the Nature of Carpels"

_plants, 2024, doi:10.3390/plants13162239_

Round 1

Reviewer 1 Report

Comments and Suggestions for Authors

To me, the fossil plant presented in this manuscript appears to be an extremely important angiosperm, because it exhibited a free-central placenta that is much more convincing than Xingxueanthus. The authors should use "free-central placenta" rather than "the seeds of Xenofructus are borne on an axis positioned between two margins." With this change, the paper may not focus on adaxial or abaxial placenta, but between marginal or central; and "carpels" can be replaced with "ovary" in some places.

Then the paper will be much more important as it presents a free-central placenta clearly.

Recommend two articles to authors:

1.      https://www.frontiersin.org/journals/plant-science/articles/10.3389/fpls.2017.01986/full

2.      https://www.frontiersin.org/articles/10.3389/fevo.2022.784077/full

Then in the Discussion, several places can be modified:

1.      “This phenomenon is so far never seen in extant angiosperms.” – How about free-central placenta”?

2.      “as it potentially represents an ancestral status of placenta.” – Why not a highly derived feature? Just describe the fossil as it is and interpret it as it should be. " Whether free-central placenta is ancestral or derived may be tested with more fossil plants to be discovered.

3.      “It is rather expected that some other primitive angiosperms may still survive in a more or less similar extant environment” can be deleted.

Comments on the Quality of English Language

There are a few places need to be revised:

1.      In abstract: “A key to these questions is older fossil fruits bearing concerned information.”— What do the authors want to say?

2.      “Fig. 2. Fruit outline is in black, while seed outline is in gray.” – How to tell black from gray on this black-white diagram? How about the central column?

3.      Replace "to" with "two" in “sandwiched between to fruit walls” 

4.      Replace "our" with "the" in “due to our lack of knowledge on Jurassic angiosperms”

5.      Should “under store of ferns” be “understory of ferns”?

Reviewer 2 Report

Comments and Suggestions for Authors

Despite the great interest in new old (pre-cretaceous) reproductive structures that may be related to the Angiosperm group, the present contribution is referred base on a poor preserved compresion material. The preservation quality is not enough to provide any microscopic level of resolution (cells nor tissues). So, authors only can interpret structures based on general morphology (not inner anatomy). I can see that even using the micro-CT sectioning (Fig. 1H-J) and 3D reconstruction (Fig. 1G) the authors can not reach enough anatomical detail in order to certificate the nature of the organs that seems to be  preserved between (sandwich-like) two larger valvate bracts. The authors interpreted these structures as seeds base on they are presented as clustered in a stalk/axis (when it may be equally interpreted as a small branch), and base on be "inside two carpels" (when they can be equally interpreted as simple bracts).

The authors interpreted the "seed" structures has a probable protrution as micropyle (Fig. 1F) but really I can see only a triangular tip in a curved leaf-like organ. There is no single character that can discern that structure is reproductive or vegetative (no, even under CT scan).

Furthermore, the authors has a circular interpretation, identifying supposedly "carpels" base on the presence of a pair of structure that "sandwiches" the supposedly seed clusters. But no offering any anatomical nor recognizable tissue in order to reinforce that circular interpretation.

The fossils in my opinion, looks like a reproductive structure (off course not as a known angiosperm group, and maybe close to a gymnosperm-like organ with small branch enclosed between two woody bracts). But authors are encourage to obtain better preserved fossils with inner anatomical details or organic-connections with other organs in order to suggest taxonomical affinities.

Round 2

Reviewer 1 Report

Comments and Suggestions for Authors

This version is much clearer. "Two alternative interpretations" and "Theoretical consideration" are very important. Well done!

Here are some minor things may need to be addressed or modified:

1. Change "The ovules are sessile, have variable orientations" to be "The ovules are sessile, with different orientations".

2. About ovules and seeds. For this fossil material how do you distinguish seeds from ovules? Fig. 3 6 has both seed and ovule, how were they determined?

3. Generally, ovules are in ovaries and seeds are in fruits. So, "ovules are supposed to be enclosed in angiosperm fruits" needs to be changed consistently.

4. "that the deployment of ovules in both fruits differ." How? Structurally? 

5. "given Shivaprakash et al. [2]." ->"given by Shivaprakash et al. [2]."

6. Spelling: "freee" should be "free"

7. Change? "numerous genes of numerous taxa" to be "numerous genes of many taxa"?

8. What do you mean by "intentional turning back to Jurassic angiosperms"?

Comments on the Quality of English Language

Very minor.

Reviewer 2 Report

Comments and Suggestions for Authors

Authors improved previous manuscript. Unfortunately the fossil preservation is very poor, so interpretations remain poorly documented. But the fossils are worth to be published.
